# Evaluation of RNA extraction and rRNA depletion protocols for RNA-Seq in eleven edible seaweed species from brown, red, and green algae

**Rob J. Dekker**, **Wim A. Ensink**, **Marina F. van Olst**, **Selina M. van Leeuwen**, **Wim C. de Leeuw**, **Martijs J. Jonker**, **Timo M. Breit**\*

RNA Biology research group, Swammerdam Institute for Life Sciences, Faculty of Science, University of Amsterdam, Amsterdam, the Netherlands

\* T.M.Breit@uva.nl

## Abstract

Seaweeds represent a promising resource for food as well as pharmaceutical and cosmetic applications. However, genomic and transcriptomic research on these organisms remains underdeveloped, partly due to technical challenges in DNA/RNA extraction and ribosomal RNA (rRNA) depletion. These challenges are particularly acute in species lacking standardized protocols. This study systematically evaluates RNA isolation and rRNA depletion protocols for RNA sequencing (RNA-seq) across 11 commercially relevant edible seaweed species, encompassing six brown (Hetero-kontophyta), four red (Rhodophyta), and one green (Chlorophyta) alga. Seven RNA extraction protocols (three CTAB-based and four spin-column-based) were compared for RNA yield, integrity, and purity. Brown seaweeds generally yielded superior RNA with CTAB-based methods, whereas red and green seaweeds performed better with chaotropic-salt-based spin-column methods. Additionally, three commercial rRNA depletion kits (Ribo-Zero Plant, riboPOOL, and RiboFree) were assessed for each seaweed species separately in a large matrix-style RNA-seq experiment (43 samples) to determine their effectiveness in removing rRNA compared to undepleted controls. Both RiboFree and riboPOOL significantly outperformed Ribo-Zero Plant, yielding post-depletion rRNA mapping rates of only 6, 9, and 19%, respectively. This study provides practical guidelines for selecting RNA isolation and ribodepletion methods tailored to specific seaweed taxa and supports the development of transcriptomic tools for seaweed research.

## Introduction

Seaweed is increasingly recognized as a sustainable and nutritionally valuable food source [1,2], as well as a rich reservoir of pharmaceutical bioactive compounds [3] and ingredients for cosmetic applications [4]. Environmental stress responses

**Data availability statement:** The RNA sequencing data generated in this study have been deposited in the NCBI Sequence Read Archive (SRA) under BioProject accession number PRJNA1276033.

**Funding:** The author(s) received no specific funding for this work.

**Competing interests:** The authors have declared that no competing interests exist.

in seaweeds have been extensively investigated [5], and advances in macroalgal genomics and transcriptomics are accelerating our understanding of their molecular biology [6]. However, in stark contrast, research on algal pathogens and associated diseases remains relatively limited, despite their substantial influence on seaweed health and aquaculture productivity [7]. The application of robust transcriptomic methodologies is therefore critical for uncovering, identifying, and characterizing seaweed pathogens, ultimately supporting the development of effective disease management strategies.

RNA sequencing (RNA-seq) has emerged as a powerful tool for studying transcriptomes, as well as detecting RNA viruses in plants and algae through metatranscriptomics [8]. By analyzing RNA profiles and sequences, RNA-seq not only enables fundamental biological research but also provides a means to diagnose viral infections, as well as identify novel RNA viruses and associated new RNA functionalities [8–10]. The success of RNA-seq, however, critically depends on the isolation of high-quality RNA, which can be particularly challenging in seaweeds due to their high polyphenol and polysaccharide content. Furthermore, the choice of library preparation method plays a critical role in capturing the full range of transcripts. While poly(A)-based selection is widely used to enrich eukaryotic mRNA, it biases against non-polyadenylated transcripts. This decreases the ability to detect non-polyadenylated endogenous transcripts as well as transcripts derived from (pathogenic/commensal) prokaryotic and viral sources. In algal systems, poly(A)-selected libraries have yielded substantially fewer viral reads and fewer viral contigs than rRNA-reduced total-RNA libraries [11]. Therefore, rRNA depletion is the preferred approach for seaweed RNA-seq [8] when comprehensive recovery of host, viral, and prokaryotic transcripts is required.

Multiple RNA isolation protocols have been optimized to overcome interference from the high polyphenol and polysaccharide content of algae [12–19]. The CTAB (cetyltrimethylammonium bromide) method is widely utilized for plant tissues with high levels of polysaccharides and polyphenolic compounds [20]. This method employs the cationic detergent CTAB in a high-salt buffer to lyse cells and selectively solubilize nucleic acids while enabling the removal of polysaccharides and contaminants. Following cell lysis, organic extraction facilitates the separation of proteins, lipids, and debris, leaving RNA-CTAB complexes in the aqueous phase. Subsequent precipitation with isopropanol or ethanol, under high salt conditions, ensures selective RNA recovery. An alternative, widely adopted method, entails the use of chaotropic agents, such as guanidine isothiocyanate, to disrupt cellular membranes and denature proteins, including RNases, ensuring RNA integrity during extraction [21,22]. RNA can then be extracted either by direct precipitation or using the Boom nucleic acid extraction method, which employs silica spin columns that selectively bind nucleic acids via ionic interactions under high-salt conditions [23]. After several washing steps, to remove contaminants while maintaining RNA binding, pure RNA is eluted with a low-salt buffer or water. There are numerous variations of these methods available in both the scientific literature and via commercially available kits. Given the vast diversity of seaweed species, developing a universal RNA extraction

protocol is likely impractical. Prior work has largely optimized RNA isolation for individual species or color groups, and one multi-taxon study evaluated only two extraction chemistries (CTAB- and SDS-based). For many species, no published high-quality RNA extraction protocols exist, forcing researchers to assess and choose the most appropriate method for their specific needs by themselves. To our knowledge, systematic comparisons of ribodepletion strategies in seaweeds have not been reported, and no commercial ribodepletion kits are currently designed specifically for algal samples.

The aim of this study was therefore to provide the first cross-taxon comparison that couples RNA isolation with ribodepletion performance for RNA-seq analysis across 11 different edible seaweed species from brown (Phaeophyceae), red (Rhodophyta), and green (Chlorophyta) algae. Specifically, we sought to identify RNA isolation protocols that yield RNA of sufficient integrity, purity, and quantity for RNA-seq and evaluate how different ribodepletion approaches remove rRNA from seaweed RNA preparations.

To achieve these objectives, we selected a representative set of RNA isolation protocols from prior publications. Seven RNA isolation protocols, including three CTAB-based methods: CTAB1 [17], CTAB2 [24], and CTAB3 [15]; and four spin-column-based methods using chaotropic salt lysis: LogSpin [25], RNeasy-M, RNeasy-P, and RNeasy-PP (specifically designed for challenging high-polysaccharide/polyphenol samples). These protocols were tested on 11 commercially relevant seaweed species, including six brown, four red, and one green species. RNA integrity, yield, and purity were assessed using spectrophotometric measurements and automated polyacrylamide gel electrophoresis. For each seaweed species, the best-performing RNA isolates were then used to evaluate the efficiency of three commercially available ribodepletion kits (Ribo-Zero Plant, riboPOOL, and RiboFree), in comparison to corresponding non-ribodepleted controls. Two kits are based on oligonucleotide hybridisation (Ribo-Zero Plant and riboPOOL) whereas RiboFree is a cross-species specific kit that uses the preferential re-hybridisation of highly abundant transcript RNAs with their cDNA counterpart, followed by selective degradation of the cDNA strand using duplex-specific nuclease [26].

This study provides a framework for selecting RNA isolation and ribodepletion protocols for transcriptomic research in seaweeds, encompassing applications from global gene expression profiling to the improved detection of known and emerging RNA viruses in cultivation systems..

## Results and discussion

In prior work, we found that RNA isolation from seaweed posed species-specific challenges. This study aimed to identify RNA isolation protocols that yield sequencing-grade RNA from a selection of commercially relevant edible seaweed species. To minimize freshness-related effects on RNA quality, we used only locally harvested material from the Netherlands, yielding 11 species across diverse taxonomic groups (S1 Table). This selection comprises a mix of model and non-model species (S1 Table) that are globally significant or regionally important within and beyond the Netherlands. [27]. Seven RNA isolation protocols (S2 Table) were evaluated in a matrix-style experiment (n = 3 per method per species). These protocols included three based on CTAB-buffer lysis followed by precipitation (CTAB1, CTAB2, CTAB3), and four based on chaotropic salt lysis followed by spin-column purification (LogSpin, RNeasy-M, RNeasy-P, RNeasy-PP). RNA integrity, yield, and purity were systematically evaluated for each combination of seaweed species and RNA isolation protocol.

### RNA yield

RNA yield was highly variable across species and protocols, spanning from 0.5 to 13 µg for all RNA extracts of sufficient quality (Table 1 and S5 Table). Overall, the CTAB1 and CTAB2 methods consistently provided the highest yields, with the exception of three red seaweed species (S03, S04, S09), where spin-column-based methods, particularly LogSpin and RNeasy-P, significantly outperformed precipitation-based methods. Notably, the CTAB3 method yielded lower RNA quantities across nearly all species, despite having a similar lysis buffer composition to CTAB1 and CTAB2. This discrepancy is likely due to the additional PCI extraction and polysaccharide precipitation steps in the CTAB3 protocol, both of which are known to reduce nucleic acid recovery.

**Table 1. RNA yield comparison across extraction methods in 11 seaweed species.**

| Sample ID | Species | RNA yield (µg) | | | | | | | | | | | | |
| | | CTAB1 | | CTAB2 | | CTAB3 | | LogSpin | | RNeasy-M | | RNeasy-P | | RNeasy-PP | |
| | | Mean | CV (%) | Mean | CV (%) | Mean | CV (%) | Mean | CV (%) | Mean | CV (%) | Mean | CV (%) | Mean | CV (%) |
| S02 | *S. muticum* | 10.2 | 79.6 | 11.0 | 32.1 | 2.4 | 141.8 | 0.3 | 36.2 | 0.8 | 21.2 | 1.0 | 6.2 | 1.3 | 118.4 |
| S07 | *A. nodosum* | 3.3 | 127.0 | 5.7 | 32.8 | 0.5 | 107.3 | 0.3 | 106.0 | 3.1 | 46.1 | 1.4 | 147.1 | 0.6 | 61.8 |
| S01 | *F. vesiculosus* | 8.4 | 66.6 | 8.9 | 13.7 | 0.4 | 30.3 | 1.1 | 48.4 | 0.9 | 48.9 | 4.1 | 5.2 | 0.6 | 90.1 |
| S06 | *F. serratus* | 6.5 | 124.8 | 8.2 | 38.6 | 0.3 | 14.1 | 0.3 | 40.0 | 1.7 | 13.8 | 1.3 | 71.3 | 0.9 | 65.6 |
| S11 | *S. latissima* | 3.1 | 134.4 | 7.8 | 25.3 | 2.0 | 109.7 | 0.8 | 18.0 | 1.7 | 38.1 | 2.1 | 42.4 | 2.4 | 48.6 |
| S10 | *D. dichotoma* | 9.6 | 71.9 | 12.6 | 42.7 | 4.1 | 122.1 | 1.7 | 21.2 | 0.3 | 15.2 | 1.3 | 57.0 | 4.1 | 50.1 |
| S08 | *C. fragile* | 2.5 | 112.8 | 5.3 | 24.4 | 2.7 | 146.2 | 4.6 | 28.4 | 5.4 | 18.7 | 4.8 | 24.6 | 3.1 | 71.1 |
| S05 | *C. crispus* | 4.8 | 86.0 | 9.5 | 46.5 | 0.9 | 86.6 | 0.5 | 14.8 | 1.8 | 31.8 | 3.2 | 23.3 | 3.0 | 108.0 |
| S04 | *G. turuturu* | 1.8 | 144.7 | 1.4 | 57.6 | 0.7 | 37.6 | 2.8 | 56.5 | 1.7 | 42.0 | 4.2 | 31.3 | 3.1 | 89.3 |
| S03 | *G. vagum* | 0.3 | 105.0 | 2.5 | 37.7 | 0.6 | 120.7 | 9.2 | 19.5 | 9.5 | 13.9 | 9.1 | 39.7 | 0.4 | 38.0 |
| S09 | *G. longissima* | 1.9 | 114.3 | 1.7 | 23.5 | 1.4 | 132.8 | 12.9 | 19.6 | 13.1 | 11.9 | 10.2 | 34.0 | 0.7 | 13.9 |

Given that all protocols were conducted starting with equal quantities (150–200 mg) from the same batch of finely ground tissue, it can be concluded that CTAB-based methods generally result in higher RNA yields; however, this increased yield did not consistently align with RNA integrity. Possible explanations include that CTAB's bulk alcohol precipitation and polysaccharide-complexing recover more total nucleic acids, including fragments, DNA, and co-contaminants, whereas silica columns are more selective but may clog or under-bind in polysaccharide-rich macroalgal lysates, reducing yield. However, additional handling before CTAB precipitation can permit RNase activity and mechanical/chemical stress, disproportionately degrading the long rRNAs used as integrity proxies. Notably, CTAB-based methods resulted in adequate RNA integrity in only five species (S01, S04, S06, S07, S11), despite higher yield overall. Spin-column-based methods, while generally yielding less RNA, offered more consistent performance for red seaweeds. Among the precipitation-based methods, CTAB1 and CTAB2 outperformed CTAB3 and most spin-column-based methods in terms of yield.

## RNA integrity

In seaweeds, abundant plastid-derived 16S and 23S rRNAs co-extract with cytosolic rRNAs, producing additional prominent peaks in electropherograms commonly used for RNA integrity assessment. These extra peaks can be misclassified by the RINe algorithm as RNA degradation, producing low or inconsistent RINe values even when cytosolic 28S/18S rRNA peaks are well-defined [15]. Accordingly, we quantified RNA integrity using the cytosolic 28S/18S rRNA peak ratio, which provided a more consistent and biologically meaningful metric; higher values (typically 1.0–1.5) reflecting higher RNA quality.

The 28S/18S rRNA ratio varied among seaweed groups and extraction protocols (Table 2 and S5 Table). For brown seaweeds, CTAB1 and CTAB2 yielded RNA with higher integrity, aligning with their better performance in yield and purity. Conversely, red and green seaweed species generally yielded RNA with better integrity when spin-column-based methods (LogSpin, RNeasy-P) were used. Although the CTAB3 method achieved acceptable integrity for nearly half of the species tested, its consistently low yield limits its practical application in most RNA-seq workflows, particularly ribodepletion-based protocols. However, this limitation could be overcome by increasing the input material.

Among the spin-column-based extraction methods, LogSpin and RNeasy-P produced comparable results in terms of yield and integrity across seaweed species. RNeasy-M offered no clear advantage in any of the seaweed species tested and was further limited by its labor-intensive procedure and reliance on hazardous phenol. As such, RNeasy-M is not recommended for RNA extraction from seaweeds in future studies.

**Table 2. RNA integrity comparison across extraction methods in 11 seaweed species.**

| Sample ID | Species | RNA integrity (28S/18S peak ratio) | | | | | | | | | | | | | PRE-FERRED METHOD* |
| | | CTAB1 | | CTAB2 | | CTAB3 | | LogSpin | | RNeasy-M | | RNeasy-P | | RNeasy-PP | | |
| | | Mean | CV (%) | Mean | CV (%) | Mean | CV (%) | Mean | CV (%) | Mean | CV (%) | Mean | CV (%) | Mean | CV (%) | |
| S02 | *S. muticum* | 0.4 | 25.0 | 0.6 | 97.2 | 0.7 | 48.2 | nd | | nd | | nd | | 0.5 | 111.4 | CTAB2 |
| S07 | *A. nodosum* | 0.9 | 87.4 | 1.0 | 36.3 | 0.3 | 173.2 | nd | | nd | | nd | | nd | | CTAB2 |
| S01 | *F. vesiculosus* | 1.0 | 34.6 | 1.1 | 46.2 | 0.5 | 89.2 | nd | | nd | | nd | | 0.7 | 110.2 | CTAB2 |
| S06 | *F. serratus* | 1.3 | 9.1 | 1.4 | 7.1 | 0.3 | 86.6 | 0.3 | 173.2 | nd | | nd | | 0.7 | 86.9 | CTAB2 |
| S11 | *S. latissima* | 1.0 | 5.6 | 1.1 | 24.1 | 1.4 | 25.8 | 1.4 | 4.2 | nd | | nd | | 0.2 | 173.2 | CTAB2 |
| S10 | *D. dichotoma* | 0.7 | 87.9 | 0.4 | 86.6 | 1.0 | 20.0 | nd | | nd | | 1 | 173.2 | 0.2 | 173.2 | CTAB1 |
| S08 | *C. fragile* | 0.5 | 43.3 | 0.3 | 88.2 | 0.7 | 24.7 | 1.2 | 14.4 | 1.2 | 4.7 | 1.3 | 4.3 | 0.3 | 86.6 | RNeasy-P |
| S05 | *C. crispus* | 0.8 | 87.5 | nd | | 1.6 | 16.5 | 1.2 | 8.3 | 0.3 | 173.2 | 1.0 | 14.8 | nd | | RNeasy-P |
| S04 | *G. turuturu* | 1.3 | 7.7 | 0.6 | 27.0 | 1.3 | 22.9 | nd | | 0 | 173.2 | 0.6 | 173.2 | 1.3 | 4.3 | RNeasy-PP |
| S03 | *G. vagum* | 0.9 | 24.0 | 0.7 | 7.9 | 1.2 | 38.3 | 1.4 | 7.1 | 0.9 | 11.1 | 1.4 | 4.0 | 0.7 | 93.7 | LogSpin |
| S09 | *G. longissima* | 0.7 | 28.4 | 0.4 | 0.0 | 0.7 | 75.6 | 1.3 | 8.7 | 0.9 | 16.4 | 1.3 | 9.1 | 0.8 | 12.5 | LogSpin |

* The preferred extraction method was determined based on combined RNA yield and integrity.

## RNA purity

RNA purity was evaluated via OD260/280 and OD260/230 absorbance ratios, which reflect contamination by proteins/phenol and salts/carbohydrates, respectively (S3, S4, and S6 Table). CTAB1 and CTAB2 consistently yielded RNA of high purity across the majority of seaweed species, likely due to the effective removal of endogenous contaminants and the exclusion of phenol and chaotropic salts from these protocols, which can otherwise persist as residues in the final RNA preparations. In contrast, methods involving phenol or chaotropic salts (LogSpin, RNeasy kits) generally resulted in suboptimal purity, particularly in brown seaweeds, possibly due to phenol and guanidinium salt carryover. However, it is important to note that purity metrics can be unreliable when RNA concentration is low. Consistent with this, RNA extracts with concentrations above 80 ng/µl consistently showed OD ratios within the optimal range, indicating high purity.

## Comparative performance of extraction chemistries across macroalgae

Across algal lineages, our RNA quality control results reflect both the cellular levels of phenolics and polysaccharides and the way each RNA extraction chemistry manages them. Brown algae are rich in phlorotannins (brown-algal polyphenols) and carry abundant alginate, fucoidan and laminarin, far exceeding the phenolic load typical of red/green taxa. In our hands, spin-column based methods therefore gave low yield, poor RNA integrity and depressed 260/280 and 260/230 ratios, consistent with (i) phenolics oxidizing to quinones that bind nucleic acids/proteins, compromising integrity and spectroscopic purity, and (ii) high molecular weight, acidic polysaccharides creating viscous, gelling lysates that hinder binding/flow and co-elute to absorb at 230 nm. Unlike other methods, CTAB2 and CTAB3 worked effectively on brown seaweeds: PVP sequesters phenolics, while high-salt CTAB followed by LiCl precipitation separates polysaccharides and ensures selective RNA precipitation, leaving DNA, proteins, and most carbohydrates in solution. This results in improved RNA integrity and OD ratios [28–31]. For red algae, whose cell walls are dominated by carrageenans and agar(ose), and for green algae, where ulvan is characteristic and phenolic levels are generally lower than in brown algae, the column-based methods tended to give higher RNA integrity values and higher total yield. The latter can be explained by the use of rapid guanidinium-salt lysis that efficiently denatures RNases (limiting RNA degradation), and CTAB protocols that commonly include LiCl precipitation, which is less efficient for very small RNAs and can reduce the total RNA mass. In these groups,

both methods delivered acceptable 260/280 and 260/230 ratios because the load of interfering polyphenols and poly-saccharides is lower than in brown seaweeds [30,32–35]. Finally, our spectrophotometric readouts align with accepted interpretations: low 260/280 indicates phenol/chaotrope/protein carryover, and low 260/230 reflects guanidinium salts and carbohydrates, precisely the contaminants expected from problematic samples and insufficient removal on spin columns when lysates are highly viscous.

## Recommended RNA extraction workflows

In the absence of consensus RNA-seq-specific quality control standards, we interpret RNA integrity and OD260/280–260/230 ratios with reference to library-prep manufacturer recommendations and general RNA purity benchmarks [36,37]. Since RNA integrity was evaluated using 28S/18S rRNA peak ratios rather than RIN values, we selected protocols with the highest ratios, applying a minimum cutoff of ≥ 1.0. Although ratios below ~1.4 are generally considered suboptimal [38], the library preparations performed on the chosen samples (see below) were not adversely affected, resulting in high-quality libraries and sequencing data.

For all species except Japanese wireweed, at least one RNA extraction method produced RNA in sufficient quantity (>1 µg), with acceptable integrity (28S/18S rRNA ratio ≥ 1.0) and optimal purity, as indicated by OD260/280 and OD260/230 ratios within the desired range. Based on these results, a single preferred RNA extraction method was selected for each seaweed species (Table 2), prioritizing RNA integrity but also considering the need for sufficient yield to support multiple downstream ribodepletion protocols. In some cases, methods offering slightly lower RNA integrity were selected due to their superior yield, ensuring adequate RNA for follow-up experiments. While these recommendations were tailored to the needs of this study, alternative applications may necessitate different choices.

For future efforts to isolate RNA from untested seaweed species, we recommend initially testing CTAB1 and/or CTAB2 for brown seaweeds, due to their consistent performance in yield, purity, and integrity. For red and green seaweed species, LogSpin and RNeasy-P are advisable starting points, providing a balance of yield and integrity with the simpler workflow typical of spin-column-based kits.

## Evaluation of ribodepletion methods for RNA-seq

For each seaweed species, we evaluated the performance of three commercially available ribodepletion kits, *i.e.,* Ribo-Zero Plant, riboPOOL, and RiboFree, by comparing them to non-ribodepleted controls. Depending on the balance between RNA quality and yield, one to three replicate RNA isolates from one sample/RNA-extraction method combination per seaweed species were selected and pooled for ribodepletion and subsequent RNA sequencing (the selected samples are highlighted in bold in S5 Table). Sequencing of these 44 samples yielded an average of 31 million reads per library (range: 17–51 million; S1 Fig).

## Residual rRNA

To estimate the residual rRNA-derived sequencing reads, we initially attempted to manually identify these sequences from *de novo* assemblies generated from non-ribodepleted controls for each species, given the limited availability of complete 28S, 18S, 5.8S, and 5S ribosomal RNA reference sequences for most seaweed species. However, reconstructing reliable full-length rRNA sequences proved infeasible without considerable additional effort, which fell outside the comparative scope of this study. As an alternative strategy, we identified the contig with the highest sequencing coverage in the non-ribodepleted control of each seaweed species and selected all contigs with coverage ≥90% of this maximum for further analysis. This lower limit was chosen because it consistently captured contigs homologous to each of the nuclear and mitochondrial rRNA species. We used this contig subset as an rRNA reference proxy and expected it to include at least all (partial) rRNAs, which are typically highly expressed and therefore exhibit high coverage in RNA-seq data [39]. This strategy can (i) inadvertently include highly expressed non-rRNA contigs, which would inflate absolute rRNA percentages but

 

affect all conditions equally, preserving relative comparisons among depletion kits within a species; and (ii) incompletely capture rRNA variants, which can lower mapping and lead to underestimation of absolute rRNA content. Accordingly, our conclusions emphasize within-species ranking of depletion methods rather than absolute rRNA percentages.

In most undepleted seaweed samples, approximately 90% (range: 71.5%–97.2%; Fig 1) of total reads aligned to the rRNA-enriched reference. Exceptions included *G. vagum* (S03), *C. crispus* (S05), *C. fragile* (S08), and *S. latissima* (S11), which showed lower mapping rates (<80%), potentially due to incompleteness of the rRNA-enriched reference because of the presence of distinct rRNA variants or inherently lower rRNA expression driven by interspecies variation, developmental stage, and/or environmental conditions. Across all seaweed species, each of the three ribodepletion kits effectively and consistently reduced the fraction of rRNA-aligned reads relative to their respective undepleted controls (Fig 1), decreasing rRNA content on average to 19% of total reads with Ribo-Zero Plant, 9% with riboPOOL, and 6% with RiboFree. Among the methods tested, the Ribo-Zero Plant kit demonstrated the lowest rRNA depletion efficiency. Coupled with its relatively complex and time-consuming protocol, it is suboptimal for transcriptomic applications in seaweed samples. In contrast, RiboFree consistently achieved the highest rRNA depletion efficiency on average, indicating its suitability for high-quality RNA-seq library preparation in these seaweed species. It is important to note that this is the only nuclease-based method included in the study and is intended to deplete highly abundant transcripts in general, rather than rRNA specifically. This broader targeting may contribute to its apparent superior performance, particularly considering that the rRNA-enriched reference proxy used for mapping predominantly represents the most highly expressed transcripts. Species-specific differences were observed: S. muticum (S02), *C. fragile* (S08), and *S. latissima* (S11) showed >2 × greater rRNA reduction with RiboFree than riboPOOL, while *G. vagum* (S03) and *G. turuturu* (S04) showed the opposite trend. However, gains in non-rRNA reads were modest (3–13%).

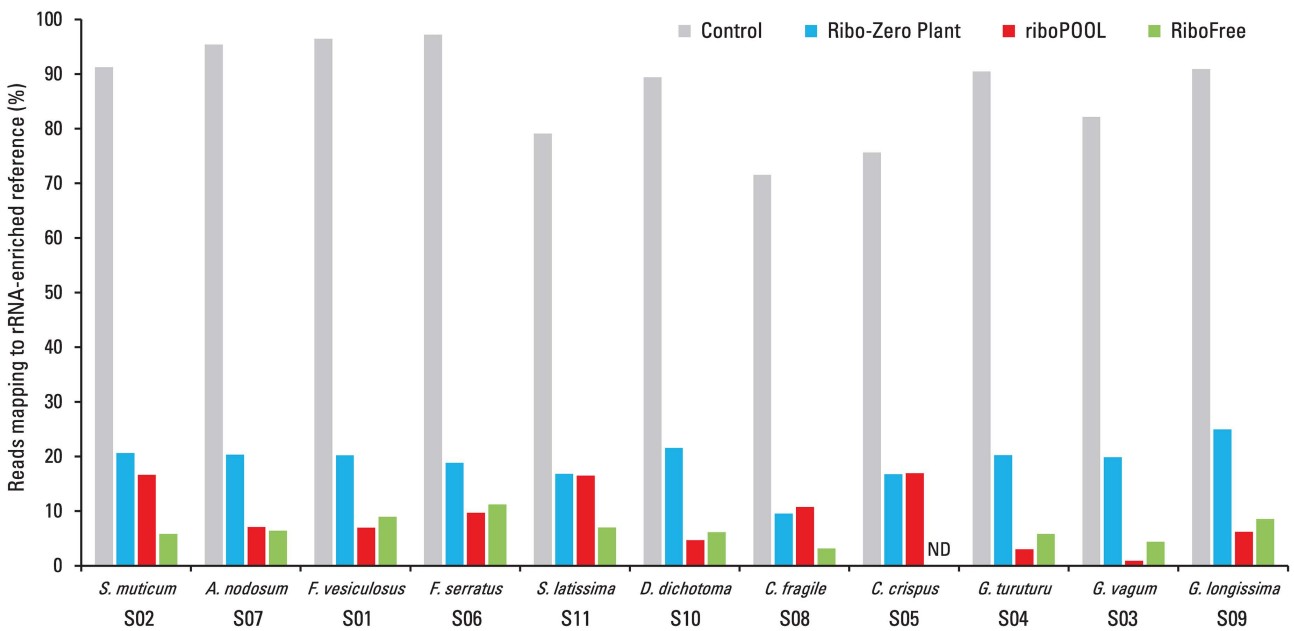

**Fig 1. Efficiency of rRNA depletion methods across 11 edible seaweed species.** The percentage of RNA-seq reads mapping to the rRNA-enriched reference in untreated (Control, grey) and ribodepleted samples using three commercial kits is shown: Ribo-Zero Plant (blue), riboPOOL (red), and RiboFree (green). Each set of bars represents a distinct seaweed species, identified by its Latin name and sample number (S01–S11; see Table 1), encompassing brown, red, and green algal groups. Sequencing library preparation from Chondrus crispus (S05) total RNA was unsuccessful for unknown reasons and is marked as "ND" (not determined).

## RNA assembly

A key objective in transcriptomic studies of newly sequenced species is the *de novo* assembly of transcripts both to construct a preliminary transcriptome and to assemble potential viral RNA genome sequences. The choice of ribodepletion protocol might significantly influence assembly quality. In this study, the ribodepletion methods were applied to the same RNA isolates for multiple seaweed species, enabling a controlled comparison of primary assembly metrics such as number of contigs and contig N50. To achieve this, RNA-seq data were downsampled separately for each seaweed species to normalize sequencing depth across the three ribodepletion methods, allowing for a more accurate comparison of assembly quality. RiboPOOL and RiboFree yielded, on average, 2.9-fold and 2.1-fold more contigs, respectively, than the Ribo-Zero Plant kit (Fig 2A). Furthermore, assemblies generated using the RiboFree data yielded a higher average contig N50 of 906 nucleotides, compared to 755 and 675 nucleotides for riboPOOL and Ribo-Zero Plant, respectively (Fig 2B). These results suggest that RiboFree may improve transcriptome assembly contiguity and increase transcript diversity, potentially by more effectively depleting rRNA and other highly abundant transcripts, and/or by preserving a broader range of transcript species.

Overall, both riboPOOL and RiboFree are effective options for rRNA depletion in seaweed, providing comparable efficiency and yielding lower residual rRNA levels than Ribo-Zero Plant.

## Cost and throughput considerations

CTAB-based precipitation uses inexpensive reagents (<$5) but requires more hands-on time, organic solvents, and occasional extra clean-ups/DNase steps, which increases labor. In contrast, spin-column protocols have higher per-sample reagent costs (~$10), but they minimize bench time and are far more user-friendly and reliable for inexperienced researchers. For rRNA depletion, commercial probe-based kits represent the main per-library expense and typically outweigh extraction costs. Approximate per-sample costs for rRNA depletion (excluding plastics, labor, and taxes) were: riboPOOL (depletion only; price shown includes an external library-prep kit),~$120; RiboFree (depletion + library prep included),~$120; and Ribo-Zero Plant (depletion + library prep included),~$160. Overall, spin-column extraction combined with either riboPOOL or RiboFree ribodepletion kits offers a practical balance between time and cost, while CTAB workflows are most cost-effective for large batches provided that labor and fume-hood availability are not limiting factors.

## Concluding remarks

Our species panel comprised six brown, four red, and one green seaweed species, reflecting local availability and commercial relevance (S1 Table). This taxonomic imbalance warrants caution when generalizing, particularly to green algae. While brown species frequently yielded higher-quality RNA with CTAB-based extractions (particularly CTAB1 and CTAB2), red (and our single green) species more often favored spin-column methods (notably LogSpin and RNeasy-P). However, RNA quality might be affected by additional factors, such as seasonal variation, disease incidence, and pollution levels. Accordingly, our conclusions are framed as starting-point recommendations, rather than universal prescriptions. Expanding the panel with additional red and especially green taxa will be important to test the robustness of these trends.

Among the ribodepletion kits evaluated, riboPOOL and RiboFree consistently outperformed Ribo-Zero Plant in reducing rRNA content. In most samples, RiboFree also lead to marginally longer contigs compared to riboPOOL. However, given the fundamentally different depletion chemistries, namely nuclease-mediated degradation in RiboFree versus oligonucleotide hybridisation in riboPOOL, method-specific biases in the representation of non-rRNA transcripts are likely and should be considered when interpreting transcriptomic profiles [40]. In addition, RiboFree is the only kit that integrates complete library preparation into its workflow, which limits compatibility with alternative library preparation protocols.

For RNA-seq in uncharacterized seaweed species, we recommend conducting initial extraction trials using CTAB1 and CTAB2 for brown algae, and LogSpin and RNeasy-P for red and green algae, rRNA depletion can then be performed

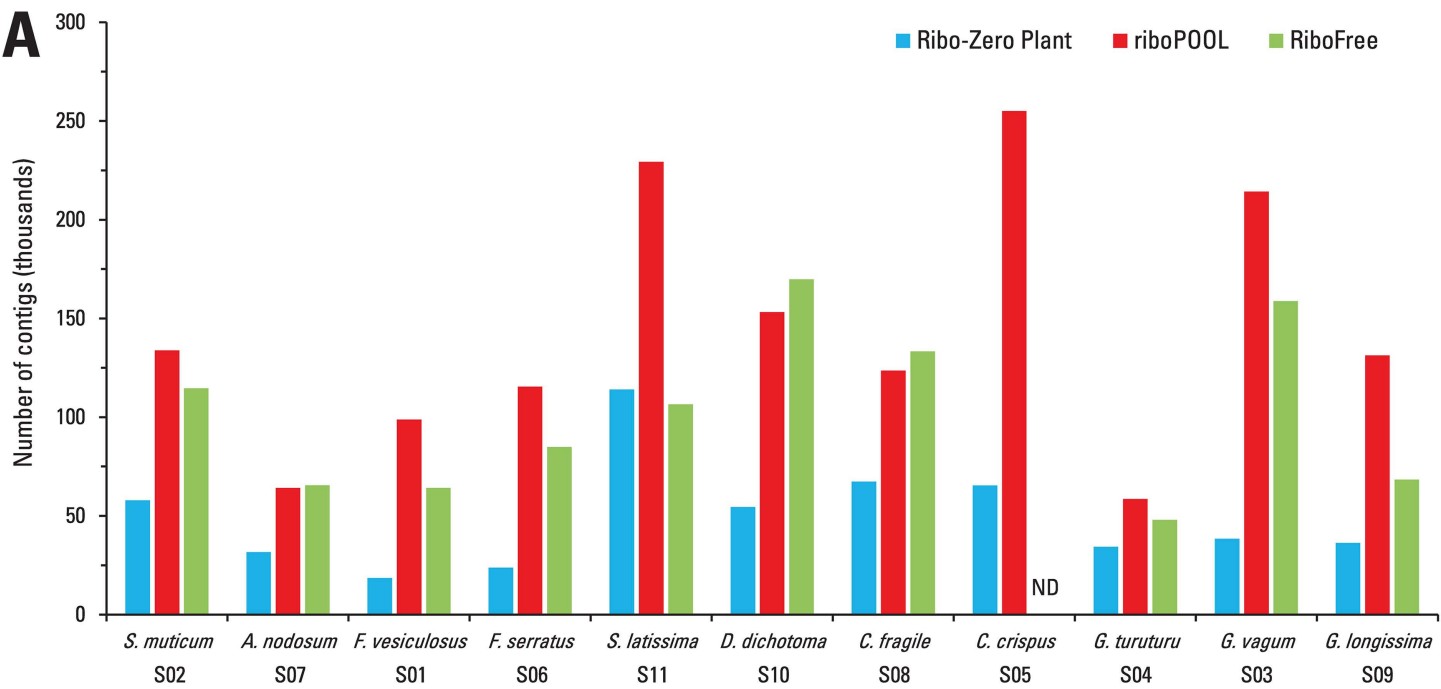

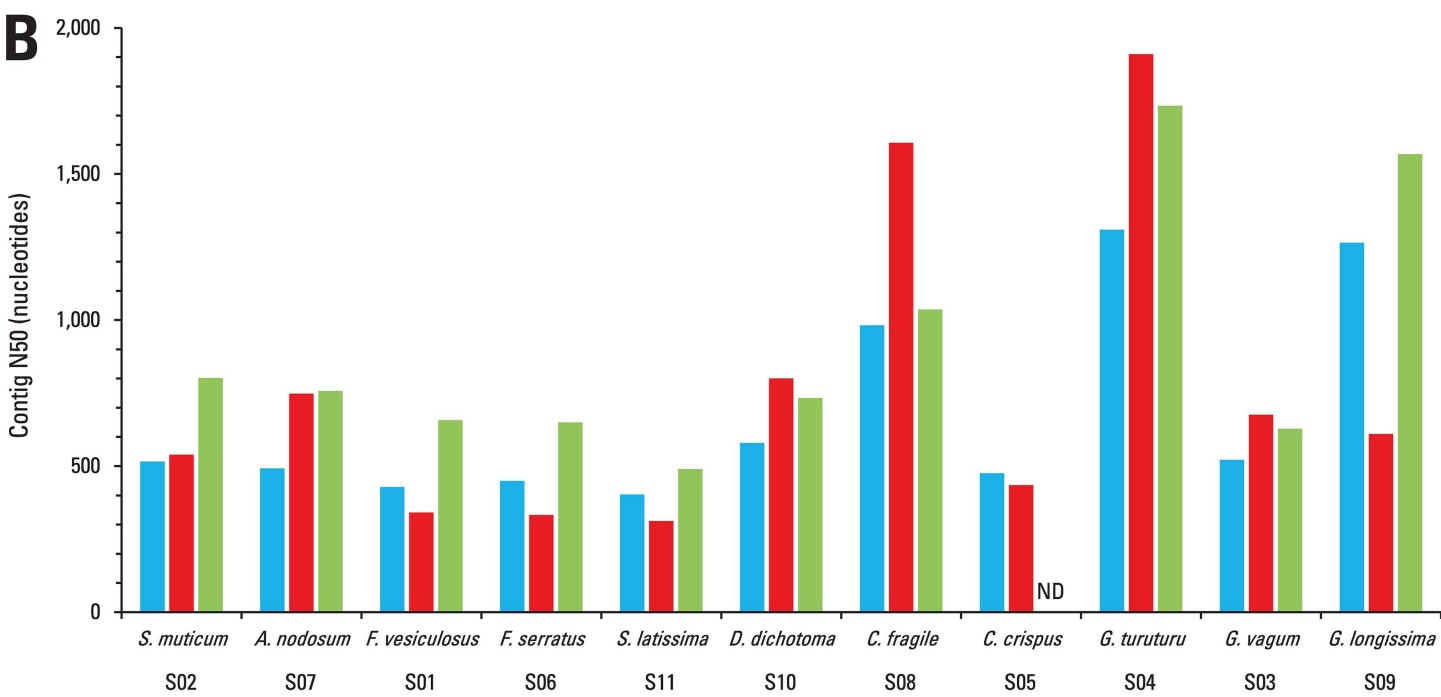

**Fig 2. Comparison of transcriptome assembly metrics across different rRNA depletion methods and seaweed species.** Before performing *de novo* transcriptome assembly, RNA-seq datasets were downsampled separately for each species to normalize sequencing depth across the three ribodepletion methods. **(A)** Total number of contigs generated following *de novo* transcriptome assembly of RNA-seq data after rRNA depletion using three methods: Ribo-Zero Plant (blue), riboPOOL (red), and RiboFree (green). **(B)** Contig N50 values for the same samples and rRNA depletion methods. Each species is denoted by its scientific name and corresponding sample ID. "ND" indicates samples for which data were not available due to failed or incomplete sequencing/assembly.

using either RiboFree or riboPOOL, based on user preference and with consideration of their distinct chemistries and compatibility requirements.

## Materials and methods

### Sample collection and preparation

Ten commercially significant edible seaweed species (S1 Table) were sourced from a small-scale local seaweed farm (Wildwier, The Netherlands), which harvests from the North Sea, the Oosterschelde estuary (tidal marine; 51°38′32.7″ N, 3°42′37.1″ E), and the Veerse Meer lake (saline lagoon; 51°32′57.1″ N, 3°52′54.0″ E), located in the southwestern region of the Netherlands (harvest date: fall 2023). Field collections were conducted under a permit issued by Rijkswaterstaat – Sea and Delta, and activities within Natura 2000 sites were authorized by the Province of Zeeland. *Saccharina latissima* was obtained from the Royal Netherlands Institute for Sea Research (NIOZ), a research facility situated in the southern Netherlands. Fresh seaweed samples were drained of excess water, flash-frozen in liquid nitrogen, and stored up to three months at –80 °C until further processing. Prior to RNA extraction, samples were finely ground under liquid nitrogen using a pre-chilled mortar and pestle to obtain a homogeneous tissue powder. Approximately 150–200 mg of ground tissue was used per extraction.

### RNA extraction methods

We first compiled publications on seaweed RNA isolation, then expanded our search to include studies focused on RNA extraction from challenging plant materials. From this pool we selected a representative set of protocols from four prior studies [15,17,24,25], excluding methods that were so similar as to be unlikely to yield different results. We also incorporated both standard and specialized column-based protocols (commercial and non-commercial) which are underrepresented in the seaweed RNA isolation literature. In total, seven RNA extraction protocols were evaluated (S2 Table). Each method was performed in triplicate for each seaweed species (n = 3 per method per species), forming a matrix-style experimental design.

CTAB1: A modified version of method 5 from Sim *et al*. was used for RNA extraction [17]. Approximately 150–200 mg of finely ground sample was added to 1 ml of extraction buffer (100 mM Tris-HCl pH 8.0, 2% (w/v) CTAB, 2 M NaCl, 20 mM EDTA, 50 mM freshly added DTT). The mixture was vortexed for 20 minutes and centrifuged at 13,000 × g for 5 minutes at room temperature. The supernatant was transferred to a fresh tube, mixed with 0.3 volumes of absolute ethanol, and briefly centrifuged. Chloroform:isoamyl alcohol (24:1, v/v) was added (200 µl per 1 ml supernatant/ethanol mixture), vortexed for 5 minutes, and centrifuged at 13,000 × g for 10 minutes. The upper aqueous phase was collected, mixed with an equal volume of 5 M LiCl (final 2.5 M), and incubated at −80 °C for ≥30 minutes. RNA was pelleted at 18,000 × g for 30 minutes at 4 °C and resuspended in 400 µl RNase-free water. A second chloroform:isoamyl extraction and LiCl precipitation were performed as above. The final RNA pellet was washed with 500 µl of 80% ethanol, air-dried, and dissolved in 20 µl RNase-free water.

CTAB2: A modified RNA extraction protocol based on Yu *et al*. was employed [24]. Approximately 150–200 mg of finely ground sample was added to 1 ml extraction buffer (100 mM Tris-HCl pH 8.0, 5% (w/v) polyvinylpyrrolidone, 3% (w/v) CTAB, 1.4 M NaCl, 20 mM EDTA, and 1% freshly added β-mercaptoethanol). The mixture was thoroughly mixed for 30 minutes, incubated at 65 °C for 10 minutes, and centrifuged at 13,000 × g for 5 minutes at room temperature. The supernatant was transferred to a fresh tube, mixed with an equal volume of chloroform:isoamyl alcohol (24:1, v/v), vigorously vortexed for 5 minutes, and centrifuged at 13,000 × g for 10 minutes. The aqueous phase was collected and extracted again with chloroform. The final aqueous layer was mixed with an equal volume of 5 M LiCl (final concentration 2.5 M) and incubated at 4 °C for 4 hours. RNA was pelleted by centrifugation at 18,000 × g for 30 minutes at 4 °C, washed twice with 75% ethanol, air-dried, and resuspended in 25 µl RNase-free water.

CTAB3: A modified RNA extraction protocol based on Jensen *et al*. was employed [15]. Approximately 150–200 mg of finely ground sample was added to 1 ml extraction buffer (100 mM Tris-HCl pH 8.0, 2% (w/v) polyvinylpyrrolidone, 2% (w/v) CTAB, 2 M NaCl, 50 mM EDTA, and 50 mM freshly added DTT) and thoroughly mixed for 30 minutes at room temperature. Phenol-chloroform extraction was carried out by adding an equal volume of ROTI Aqua-P/C/I (Carl Roth), followed by 5 minutes of vigorous vortexing and centrifugation at 13,000 × g for 10 minutes. The aqueous (upper) phase was carefully removed and transferred to a new tube, mixed with 0.3 volumes of absolute ethanol, incubated at room temperature for 5 minutes, and centrifuged again at 13,000 × g for 10 minutes. The supernatant was subjected to a second phenol-chloroform extraction. The resulting aqueous phase was mixed with 1 volume of 5 M LiCl (final 2.5 M) and incubated overnight at 4 °C. RNA was pelleted by centrifugation at 18,000 × g for 30 minutes at 4 °C, washed with −20 °C 80% ethanol, and centrifuged again for 10 minutes. The pellet was air-dried and resuspended in 20 µl RNase-free water.

LogSpin: A modified RNA extraction protocol based on Yaffe *et al*. was employed [25]. Approximately 150–200 mg of finely ground sample was added to 0.5 ml extraction buffer (8 M guanidine hydrochloride, 20 mM MES hydrate, 20 mM EDTA) and vigorously vortexed for 1 minute. After centrifugation at 15,000 × g for 5 minutes at room temperature, the clear supernatant was transferred to a new tube, mixed with an equal volume of absolute ethanol, and briefly vortexed. The mixture was applied to a spin column (E.Z.N.A. MicroElute RNA Clean Up Kit, Omega Bio-tek) and centrifuged at 8,000 × g for 1 minute. The column was washed with 400 µl of 3 M sodium acetate, followed by two washes with 500 µl of 70% ethanol, each with centrifugation at 8,000 × g for 1 minute. The column was then transferred to a clean tube and centrifuged at 13,000 × g for 5 minutes to remove residual wash buffer. RNA was eluted in 20 µl RNase-free water and collected by centrifugation at 13,000 × g for 1 minute.

RNeasy-M: Total RNA was extracted using the RNeasy Micro Kit (Qiagen), in accordance with the manufacturer's protocols outlined in Appendices C and D of the RNeasy Micro Handbook (HB-1920–003, 03/2021). Briefly, approximately 150–200 mg of finely ground sample was homogenized in 1 ml of QIAzol Lysis Reagent (Qiagen). Following phase separation, RNA was extracted from the aqueous phase using silica membrane spin columns. To ensure removal of genomic DNA, on-column DNase digestion was performed as described in Appendix D. The purified RNA was eluted in 16 µl of RNase-free water.

RNeasy-P: Total RNA was extracted using the RNeasy Plant Mini Kit (Qiagen), following the manufacturer's protocol titled "Purification of Total RNA from Plant Cells and Tissues and Filamentous Fungi", as detailed in the RNeasy Mini Handbook (HB-0435–007, 06/2023).

RNeasy-PP: Total RNA was extracted using the RNeasy PowerPlant Kit (Qiagen), according to the manufacturer's protocol provided in the RNeasy PowerPlant Kit Handbook (HB-2264–001, 10/2017).

## RNA quantification and quality assessment

RNA yield and purity were assessed by spectrophotometric analysis using a NanoDrop 2000 (Thermo Fisher Scientific). Absorbance ratios at 260/280 nm and 260/230 nm were used to evaluate contamination by protein/phenol and polysaccharides/salts, respectively.

RNA integrity was initially evaluated using the 2200 TapeStation System with RNA ScreenTapes (Agilent). However, the RNA Integrity Number equivalent (RINe) values were deemed unreliable due to interference from plastid-derived rRNA. As a result, RNA integrity was assessed based on the 28S/18S rRNA ratio, calculated from TapeStation electropherograms using the peak molarity (nmol/l) of the 28S and 18S rRNA peaks as quantified by the manufacturer's software (TapeStation Analysis Software A.02.02/SR1; Agilent Technologies). A 28S/18S rRNA ratio in the range of 1.0–1.5 was considered indicative of high-quality RNA suitable for downstream molecular applications.

For each seaweed species and RNA isolation method, RNA was independently extracted in triplicate from the same batch of finely ground sample. Mean values for the OD 260/280 and OD 260/230 absorbance ratios, as well as the

28S/18S rRNA ratio, were calculated. The coefficient of variation (CV%) was computed to assess the reproducibility of the measurements.

## Ribodepletion and RNA-sequencing

Total RNA (200–250 ng) was subjected to ribosomal RNA depletion and subsequent RNA sequencing using three distinct methods, following the manufacturer's protocols for each respective kit. For the *Ribo-Zero* method, 200 ng of total RNA was processed using the TruSeq Stranded Total RNA with Ribo-Zero Plant kit (Illumina) along with TruSeq RNA UD Indexes v2 (Illumina). For the *RiboPOOL* method, 200 ng of total RNA underwent ribodepletion using the Seawater riboPOOL kit (siTOOLs Biotech), followed by library preparation using the NEBNext Ultra II Directional RNA Library Prep Kit for Illumina (E7760; New England Biolabs) with NEBNext Multiplex Oligos for Illumina – 96 Unique Dual Index Primer Pairs (E6440; New England Biolabs). The RiboFree method involved ribodepletion and RNA-seq library preparation from 250 ng of total RNA using the Zymo-Seq RiboFree Total RNA Library Kit (R3000; Zymo Research) with the Zymo-Seq UDI Primer Set (D3008; Zymo Research). For the *non-ribodepleted control* samples, 100 ng of total RNA was directly used for library preparation with the NEBNext Ultra II Directional RNA Library Prep Kit (E7760) and NEBNext Multiplex Oligos (E6440). All resulting libraries were assessed for quality using a 2200 TapeStation System with D1000 or D5000 ScreenTapes (Agilent) and quantified using the NEBNext Library Quant Kit for Illumina (E7630; New England Biolabs). Pooled libraries were sequenced on an Illumina NovaSeq X Plus platform and base calling and quality control were performed using Illumina BclConvert v4.3.6.

## Data analysis

Sequencing data were processed through a bioinformatics workflow for quality control, assembly, contig selection, clustering, and read quantification. First, raw paired-end reads were trimmed using Trimmomatic v0.39 [41] with parameters PE, -phred33, ILLUMINACLIP:adapters:2:30:10, and MINLEN:36, discarding reads shorter than 36 bp.

For non-ribodepleted samples, trimmed paired-end reads were assembled *de novo* using SPAdes v3.15.5 [42] with parameters --cov-cutoff 10 and --rna, resulting in an average of 21,553 contigs per sample (range: 6,016–45,075). High-coverage contigs, defined as those with coverage exceeding 90% of the highest coverage contig within each sample, were selected using coverage information encoded in the fasta headers by SPAdes, resulting in on average 227 contigs per sample (range: 144–321). Contigs were clustered at 95% identity using CD-HIT v4.8.1 [43] with parameter -c 0.95 to reduce redundancy.

For each seaweed species individually, trimmed paired and unpaired reads from the ribodepletion-treated samples and non-depleted control were aligned to the corresponding set of clustered high-coverage contigs using Bowtie2 v2.4.1 [44] with parameter --very-sensitive. Reads with one or both mates aligning to the high-coverage contigs were counted using SAMtools v1.9 [45]. Total read counts were determined from trimmed data by counting paired reads as one and including unpaired R1 and R2 reads, allowing normalization of mapped read counts per sample.

To assess the number of contigs and contig N50 from the de novo assembly of ribodepleted samples, RNA-seq reads were downsampled to equalize sequencing depth across samples within each species enabling a more accurate comparison of assembly quality across different ribodepletion methods. Downsampling was performed using seqtk v1.3 (https://github.com/lh3/seqtk), retaining the following number of reads (in millions) per sample: 22 (S01), 19 (S02), 25 (S03), 17 (S04), 19 (S05), 15 (S06), 16 (S07), 24 (S08), 21 (S09), 24 (S10), and 16 (S11).

## Supporting information

**S1 Fig. Total number of trimmed RNA-seq reads obtained from 11 seaweed species using four ribodepletion methods.** Bar plots represent the total number of trimmed reads for each sample across four ribodepletion protocols: Control (no depletion, gray), Ribo-Zero Plant (blue), riboPOOL (red), and RiboFree (green). Each set of bars represents a

distinct seaweed species, identified by its Latin name and sample number (S01–S11; see Table 1), encompassing brown, red, and green algal groups. Sequencing library preparation from *Chondrus crispus* (S05) total RNA was unsuccessful for unknown reasons and is marked as "ND" (not determined).
(PDF)

**S1 Table. Taxonomic identification of seaweed species used in this study.**
(XLSX)

**S2 Table. Summary of RNA extraction protocols.**
(XLSX)

**S3 Table. RNA purity (OD260/280) across extraction methods.**
(XLSX)

**S4 Table. RNA purity (OD260/230) across extraction methods.**
(XLSX)

**S5 Table. RNA yield and integrity values for individual replicates.**
(XLSX)

**S6 Table. OD260/280 and OD260/230 values for individual replicates.**
(XLSX)

## Acknowledgments

We thank Klaas R. Timmermans of the Royal Netherlands Institute for Sea Research (NIOZ) for generously providing samples of *Saccharina latissima*.

## Author contributions

**Conceptualization:** Rob J. Dekker, Wim A. Ensink, Timo M. Breit.

**Formal analysis:** Wim C. de Leeuw.

**Investigation:** Wim A. Ensink, Marina F. van Olst, Selina M. van Leeuwen.

**Project administration:** Selina M. van Leeuwen.

**Software:** Wim C. de Leeuw.

**Supervision:** Rob J. Dekker, Martijs J. Jonker, Timo M. Breit.

**Visualization:** Rob J. Dekker.

**Writing – original draft:** Rob J. Dekker.

**Writing – review & editing:** Rob J. Dekker, Martijs J. Jonker, Timo M. Breit.

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
