## [Decision Letter · Decision Letter 0]

11 Sep 2025

Dear Dr. Rob J. Dekker,

Thank you for submitting your manuscript to PLOS ONE. After careful consideration, we feel that it has merit but does not fully meet PLOS ONE’s publication criteria as it currently stands. Therefore, we invite you to submit a revised version of the manuscript that addresses the points raised during the review process.

Your study provides a timely and useful contribution by systematically evaluating RNA extraction and rRNA depletion methods in multiple edible seaweed species. The reviewers appreciate the technical design and the practical value of your findings. However, each reviewer has identified areas where clarification, additional detail, and refinement are required to strengthen the manuscript.

Taken together, the reviews acknowledge that your manuscript addresses an important methodological gap in algal transcriptomics and that the experimental design is sound. At the same time, the reviewers request additional methodological details, clarification of sampling choices, more careful discussion of results, and improvements in presentation.

In light of these comments, the editorial decision is Major Revision. We encourage you to revise the manuscript by addressing each reviewer’s comments thoroughly, incorporating missing methodological details, strengthening the discussion with biochemical and practical insights, and improving figure and text clarity. A carefully revised version will be eligible for further consideration.

We appreciate the effort you have invested in this work and look forward to receiving your revised submission.

We look forward to receiving your revised manuscript.

Kind regards,

Inbakandan Dhinakarasamy, Ph.D

Academic Editor

PLOS ONE

Journal Requirements:

4. We are unable to open your Supporting Information file [S1 Fig]. Please kindly revise as necessary and re-upload.

Additional Editor Comments (if provided):

Reviewer #1: found the study to be well structured but noted that some parts of the Introduction contain results, which should be moved to the appropriate section. They also highlighted that the mention of RNA viruses requires clearer context. Concerns were raised about the taxonomic imbalance of your sampling, with six brown algae dominating compared to four red and one green alga, which could bias the conclusions. This limitation should be discussed explicitly. Reviewer 1 also emphasized the need for more rigorous statistical treatment of your data, for biochemical explanations for poor yields in some species, and for clarification of the discrepancy in starting material (150 µg vs. 150 mg). Importantly, they encouraged you to align your reported RNA quality metrics with accepted RNA-seq thresholds and to provide practical considerations such as cost, time, and ease of use for each method, which would greatly enhance the utility of your work.

Reviewer #2: praised the overall design but requested several clarifications in the Introduction and Methods. They suggested specifying which environmental stressors have already been studied in seaweeds, and citing evidence of viral transcripts being missed in polyA-based RNA-seq approaches. They also asked you to clearly state whether this is the first comparative evaluation in seaweeds. In the Methods, they recommended more transparency on RNA integrity after frozen storage, measurement of polysaccharide/polyphenol carryover in CTAB methods, use of RNase inhibitors in the LogSpin method, and justification for the 90% coverage threshold used. In the Discussion, they encouraged you to explain why CTAB methods yielded more RNA but lower integrity, why spin-column methods worked better for red algae, and how suboptimal purity was validated in brown algae. A more detailed interpretation of these biochemical differences would considerably strengthen your conclusions.

Reviewer #3: raised points regarding species selection and contextual information. They asked you to clarify why the eleven species were chosen and whether they are commercially significant. The reviewer also requested that latitude, longitude, and environmental parameters of the collection sites be included. They suggested you justify your choice of CTAB and spin-column protocol variants given the range of options available. The reviewer also recommended that you briefly explain how plastid-derived rRNA interferes with RINe scoring in algae, and that you acknowledge the limitations of incomplete rRNA assemblies in your study.

Reviewer #4: provided several constructive suggestions for improving clarity and presentation. They recommended including genus names and phyla in the Abstract, modifying the phrasing in Line 31 to “environmental stress factors,” and relocating Lines 78–84 of the Introduction to the Results/Discussion. They also suggested including NanoDrop quantification in the methodology to support RNA purity measurements, refining the section title at Line 144, and more clearly justifying the novelty of the work. Figures 1 and 2 should be revised to include standard errors for completeness.

Reviewers' comments:

Reviewer's Responses to Questions

**Comments to the Author**

1. Is the manuscript technically sound, and do the data support the conclusions?

Reviewer #1: Yes

Reviewer #2: Yes

Reviewer #3: Partly

Reviewer #4: Yes

2. Has the statistical analysis been performed appropriately and rigorously?

Reviewer #1: No

Reviewer #2: No

Reviewer #3: Yes

Reviewer #4: No

3. Have the authors made all data underlying the findings in their manuscript fully available?

Reviewer #1: Yes

Reviewer #2: Yes

Reviewer #3: Yes

Reviewer #4: No

4. Is the manuscript presented in an intelligible fashion and written in standard English?

Reviewer #1: Yes

Reviewer #2: Yes

Reviewer #3: Yes

Reviewer #4: Yes

Reviewer #1: The manuscript “Evaluation of RNA extraction and rRNA depletion protocols for RNA-Seq in eleven edible seaweed species from brown, red, and green algae” presents a well-designed comparative evaluation of RNA extraction and rRNA depletion protocols across a taxonomically diverse set of edible seaweed species. The study addresses important technical aspects in algal transcriptomics and provides practical recommendations for future research. However, I have some suggestions and clarification details that authors have to include in the manuscript.

1. Is it clear why these specific 11 seaweed species were selected, and are they commercially significant varieties?

2. Collection area: Latitude and longitude should be included in the manuscript

3. On what criteria did the authors choose the protocols (RNA isolation) used in this study? The choice of specific CTAB and spin-column protocol variants should be justified, as numerous alternatives exist in the literature.

4. Include the relevant environmental parameters of the sample collection area.

5. The use of the 28S/18S ratio over RINe values is appropriate; however, a brief explanation of how plastid-derived rRNA interferes with RINe scoring in algae would improve accessibility for readers outside the field.

6. The limitations of this method (constructing an rRNA), particularly with incomplete rRNA assemblies, should be acknowledged in the discussion.

Reviewer #2: Reviewer comments

The manuscript titled “Evaluation of RNA extraction and rRNA depletion protocols for RNA-Seq in eleven edible seaweed species from brown, red, and green algae (Manuscript No. PONE-D-25-34674)” provides a valuable cross-species methodological comparison. At the same time, I feel there is no need of this paper altogether, as literature on RNA extraction for various seaweeds are already available.

Coming to the article, the study is well-structured and offers practical guidance for researchers working on non-model algal species (though there are hundreds of seaweeds species exists, and studies have been done only on 11 including 6 brown algae). However, several aspects require further clarification and refinement to ensure reproducibility, transparency, and broader applicability. Please address the following comments.

Comments

1. The Introduction section contains results (lines 77–84) that should be removed to maintain focus on the background and study rationale. Point on RNA virus in the first para of introduction is a misfit. If you want to emphasize on RNA viruses, start with a fresh paragraph.

2. The study evaluates RNA isolation and rRNA depletion protocols across 11 seaweed species; however, the sampling is skewed toward brown algae (6 species) compared to red algae (4 species) and green algae (1 species). This taxonomic imbalance could bias conclusions toward protocols that perform better for brown algae. Please discuss this limitation and whether the observed trends are likely to hold across a more balanced species representation.

3. To strengthen the evaluation of RNA isolation and rRNA depletion protocols, we recommend applying a more rigorous statistical framework.

4. The discussion could be strengthened by hypothesizing why certain species (especially some red algae) yield poor RNA with specific methods, referencing known polysaccharide or polyphenol interference.

5. In result, starting material used is written as 150 microgram and in method its 150 mg. Which is correct?

6. The manuscript reports RIN values and absorbance ratios for RNA quality assessment but does not discuss whether these meet accepted RNA-seq input quality thresholds. Please address how the measured values align with recommended standards, and if any deviations occurred, discuss potential implications for downstream RNA-seq performance.

7. While the manuscript provides a robust technical comparison of RNA isolation and rRNA depletion protocols, it would benefit from including practical considerations such as approximate cost per sample, processing time, and ease of use for each method. This information would make the conclusions more actionable, particularly for laboratories with limited budgets or resources.

Reviewer #3: Reviewer Comments

Overall, the manuscript addresses an important methodological framework in algal transcriptomics by systematically evaluating RNA extraction and rRNA depletion protocols across a diverse set of seaweed species. The manuscript is generally well structured and the experimental design appears rigorous. However, there are areas where clarifications are needed before the manuscript can be accepted.

Introduction

Pg. 2, Line No.30-32: The authors stated that “advancing our understanding of seaweed biology, including its response to environmental stressors and pathogens, is essential for enhancing cultivation practices.” Please clarify which stressors (temperature, salinity, pollutants) have already been studied in seaweeds, and which remain underexplored?

Pg. 2, Line No.39-44: The authors have mentioned “RNA-seq limitations in capturing non-polyadenylated RNAs. Since many algal viruses and prokaryotic RNAs lack polyA tails, could the authors provide evidence from prior algal/seaweed transcriptome studies where viral transcripts were missed due to polyA-based methods?

Pg. 2, Line No.45–61: Have there been any comparative evaluations of these methods specifically in seaweeds before? If not, please make clear that this study is addressing the first comparison report.

Methods

Pg. 11, Line No. 245–251: The authors mentioned flash-freezing in liquid nitrogen and storage at

-80 °C. Since long-term frozen storage can sometimes affect RNA quality in high-polysaccharide tissues, was RNA integrity assessed immediately after storage to ensure no degradation during sample handling?

In CTAB methods, do authors measured the carryover of polysaccharides or polyphenols (e.g., by A230 contamination ratios or carbohydrate assays) to confirm the effectiveness of CTAB-based removal?

In LogSpin method, any RNase inhibitor was added during extraction, since guanidine-based buffers can still leave trace RNase activity in seaweed tissues?

Pg. 15, Line No.350–353, the cutoff of 90% coverage relative to the highest coverage contig was used. How was this threshold determined?

Results and Discussion

Page 5, Line No.108–110: CTAB-based methods generally yielded higher RNA quantities but did not consistently align with acceptable RNA integrity. What underlying molecular or biochemical factors could cause this discrepancy? Please discuss.

Page 5, Line No 111–113: Spin-column methods performed better in red seaweeds. What specific traits of red seaweeds influenced their compatibility with spin-column methods compared to CTAB-based ones?

Page 6, Line No 135-141: Methods involving phenol/chaotropic salts showed suboptimal purity in brown seaweeds. how was this validated? as residual contaminants directly affect downstream ribodepletion or sequencing efficiency.

Overall, the authors need to provide a more detailed discussion of the results obtained.

Reviewer #4: Reviewer comments

Abstract

The genus names of Seaweeds has to be mentioned for more clarity along with the phylum name. (Line nos 17 & 18)

Introduction

In Line no. 31 Include as “Environmental Stress factors….”

Line nos 78 – 84 can be removed from Introduction part and may be included in Results and discussion column.

Results and Discussion

Nano drop quantification may be included in the methodology for the justification of purity of RNA.

Line no 144 Title has to be precise and meaningful.

Kindly justify the novelty of the work.

Fig 1 & 2 Standard errors may be provided in the figures.

**Do you want your identity to be public for this peer review?** For information about this choice, including consent withdrawal, please see our Privacy Policy

Reviewer #1: **Yes: ** T. Stalin Dhas, Centre for Ocean Research, Sathyabama Institute of Science and Technology

Reviewer #2: No

Reviewer #3: **Yes: ** Dr. Jeyapragash Danaraj

Reviewer #4: No

---

## [Author Response · Author response to Decision Letter 1]

4 Nov 2025

Reviewer #1

The manuscript “Evaluation of RNA extraction and rRNA depletion protocols for RNA-Seq in eleven edible seaweed species from brown, red, and green algae” presents a well-designed comparative evaluation of RNA extraction and rRNA depletion protocols across a taxonomically diverse set of edible seaweed species. The study addresses important technical aspects in algal transcriptomics and provides practical recommendations for future research. However, I have some suggestions and clarification details that authors have to include in the manuscript.

We sincerely appreciate the reviewer’s time and effort in carefully evaluating our manuscript, and for providing valuable suggestions and corrections that have greatly enhanced its quality.

1. Is it clear why these specific 11 seaweed species were selected, and are they commercially significant varieties?

Since using fresh material is essential to minimize effects on RNA quality, we made an effort to obtain locally harvested seaweed in the Netherlands, which is limited due to permit requirements. The species we selected include all those currently harvested by the permit holder (WildWier). These species are mainly collected for food use in the Netherlands, but many are also of global or regional importance outside of the Netherlands (see the FAO report, The Global Status of Seaweed Production, Trade and Utilization). To clarify our sampling rationale, we have added a brief explanation in the Results section and included this report as a reference (L93–98).

2. Collection area: Latitude and longitude should be included in the manuscript

Agreed. We have added site coordinates to the ‘Sample collection and preparation’ section in the Materials and Methods (L303-306). Saccharina latissima was obtained from NIOZ (research facility supply, not a field collection), so no field coordinates apply for that entry.

3. On what criteria did the authors choose the protocols (RNA isolation) used in this study? The choice of specific CTAB and spin-column protocol variants should be justified, as numerous alternatives exist in the literature.

We agree that the manuscript would benefit from making our selection rationale more explicit. Indeed many protocols exist, but these have by far mostly been developed for non-algae. At the start of this study, which was not originally designed to tackle RNA isolation issues, we have scanned literature for protocols specifically addressing RNA isolation from macro-algae, including one protocol that is aimed at isolating RNA from strawberry, which is plagued by high polysaccharide/polyphenol levels. The other class of RNA isolation protocols, i.e. based on chaotropic-salt lysis and silica spin columns, is used extensively for many species and tissue types. Among these, the RNeasy PowerPlant kit was specifically designed for plant samples with high polysaccharide and polyphenol content. We included further information on the selection of RNA isolation methods for this study (L315-320).

4. Include the relevant environmental parameters of the sample collection area.

Because this study focused on optimizing RNA isolation and ribodepletion protocols rather than testing biological hypotheses, we unfortunately did not collect environmental parameters. The only contextual information we can still obtain at this time is the classification of collection sites as tidal marine or saline lagoon, including the exact location coordinates. We have added this to the Materials and Methods (L303-308).

5. The use of the 28S/18S ratio over RINe values is appropriate; however, a brief explanation of how plastid-derived rRNA interferes with RINe scoring in algae would improve accessibility for readers outside the field.

We agree and have added an improved explanation (L128-133) to clarify why RINe is unreliable in seaweeds.

6. The limitations of this method (constructing an rRNA), particularly with incomplete rRNA assemblies, should be acknowledged in the discussion.

We appreciate the reviewer’s point. In the original manuscript we discussed only one limitation of our rRNA proxy, i.e. false positives arising when some non-rRNA reads map to rRNA references. We agree that the complementary issue, incomplete coverage of the full set of nuclear, plastid, and mitochondrial rRNAs leading to false negatives, was not explicitly addressed. We have now added text in the discussion section (L213-221) noting that such incompleteness could cause underestimation of absolute rRNA fractions and that these values should be interpreted as lower bounds. Importantly, the same proxy and reference set were applied uniformly to all samples, including the no-depletion controls, so any residual bias is largely systematic across conditions. Because our inferences rely on relative comparisons among methods rather than absolute percentages, this limitation does not change our main conclusions.

Reviewer #2:

The manuscript titled “Evaluation of RNA extraction and rRNA depletion protocols for RNA-Seq in eleven edible seaweed species from brown, red, and green algae (Manuscript No. PONE-D-25-34674)” provides a valuable cross-species methodological comparison. At the same time, I feel there is no need of this paper altogether, as literature on RNA extraction for various seaweeds are already available.

Coming to the article, the study is well-structured and offers practical guidance for researchers working on non-model algal species (though there are hundreds of seaweeds species exists, and studies have been done only on 11 including 6 brown algae). However, several aspects require further clarification and refinement to ensure reproducibility, transparency, and broader applicability. Please address the following comments.

We thank the reviewer for their constructive evaluation and for highlighting the strengths of our work. We appreciate the concern regarding the paper’s necessity. Although multiple protocols for RNA isolation from seaweeds have been published (refs. 10, 12–18), we encountered substantial challenges across the 11 species examined, and selecting an optimal method was not straightforward. Accordingly, our goal was to provide a cross-species comparison that includes a representative selection of methods reported in previous studies. Furthermore, to our knowledge, no commercial rRNA-depletion tools are specifically designed for macroalgae; a systematic evaluation of general-purpose ribodepletion protocols in this context has been lacking. The rationale for our study has now been expanded (L63-74), thereby also addressing the concern raised by reviewer #3.

1. The Introduction section contains results (lines 77–84) that should be removed to maintain focus on the background and study rationale. Point on RNA virus in the first para of introduction is a misfit. If you want to emphasize on RNA viruses, start with a fresh paragraph.

We thank the reviewer for this helpful comment. In some research areas with which we are more familiar, it is common practice to summarize key results and conclusions at the end of the introduction. However, we now recognize that in this field such information is not typically included in the introduction. We have therefore removed the results from lines 77–84. In addition, we judged that elaborating on RNA viruses in a separate paragraph would not align with the main focus of this paper and thus removed the sentence.

2. The study evaluates RNA isolation and rRNA depletion protocols across 11 seaweed species; however, the sampling is skewed toward brown algae (6 species) compared to red algae (4 species) and green algae (1 species). This taxonomic imbalance could bias conclusions toward protocols that perform better for brown algae. Please discuss this limitation and whether the observed trends are likely to hold across a more balanced species representation.

We agree that our sampling is weighted toward brown algae (6 brown, 4 red, 1 green) and that this affects how broadly results can be generalized. This limitation is now explicitly acknowledged, emphasizing that although results observed within a particular colour group may hold for the taxa examined, they cannot necessarily be generalized to all species within that group (L283-289).

3. To strengthen the evaluation of RNA isolation and rRNA depletion protocols, we recommend applying a more rigorous statistical framework.

We agree with the reviewer, as we also preferred to apply a rigorous statistical frame work, but after consulting with our biostatistician during the course of this project, we came to the conclusion that this is not readily implementable for the following reasons:

- For our main QC readouts (RNA integrity, yield, OD260/230, and OD260/280) there are, to our knowledge, no de facto standard statistical methods available.

- We have three technical replicates per condition, and they’re noisy. This matches our experience: “easy” samples give very consistent QC, while difficult tissues vary more even when treated the same way. Assessing outlier-behaviour of samples is in this context based on laboratory experience. In order to give maximum transparency to the readers, we have therefore provided all individual QC values in S5 and S6 Table.

- For the final advice for which protocol to use on which species, we have given different priorities to different QC metrics (as described in L189-194). This is difficult to take into account in a formal statistical test and has limited added value. Readers can interpret the results by assigning their own relative importance to the various QC metrics and select the method that best aligns with their specific goals. For example, if large quantities of material are required for multiple downstream assays, yield may be prioritized over integrity or purity. A similar study by Jensen et al. ([15] in our paper) used the same metrics and also did not apply statistical tests.

Our goal was practical: identify the protocol that is most likely to work across challenging samples, and give the reader insight we use descriptive statistics rather than inferential.

4. The discussion could be strengthened by hypothesizing why certain species (especially some red algae) yield poor RNA with specific methods, referencing known polysaccharide or polyphenol interference.

Our original aim was to keep the paper focused on the practical selection of working protocols; however, we now recognize that a more extensive discussion of the results would strengthen the manuscript. We have therefore added a biochemistry-focused paragraph to the results/discussion that more extensively discusses our results (L157-179).

5. In result, starting material used is written as 150 microgram and in method its 150 mg. Which is correct?

Thank you for catching this error: 'microgram' should have been 'milligram' (L113).

6. The manuscript reports RIN values and absorbance ratios for RNA quality assessment but does not discuss whether these meet accepted RNA-seq input quality thresholds. Please address how the measured values align with recommended standards, and if any deviations occurred, discuss potential implications for downstream RNA-seq performance.

To our knowledge, there is no single official standard for RNA QC metrics accepted for RNA-seq. Most recommendations for minimum RNA yield and integrity are provided in the manuals of library preparation kits supplied by manufacturers. OD260/280 and 260/230 ratios serve as more general indicators of RNA purity used for different applications. Several studies suggest broadly similar QC requirements that apply to gene expression analyses in general. We have included a brief discussion of this point and cited the relevant references to inform readers that guidelines on this matter exist (L181-186).

7. While the manuscript provides a robust technical comparison of RNA isolation and rRNA depletion protocols, it would benefit from including practical considerations such as approximate cost per sample, processing time, and ease of use for each method. This information would make the conclusions more actionable, particularly for laboratories with limited budgets or resources.

Thank you for this suggestion. We agree that cost information is valuable for readers. We have added a paragraph discussing the costs of the different protocols and kits, along with a rough comparison of their labour requirements (L263-273), to help readers quickly determine which approaches are generally more economical. 

Reviewer #3

Overall, the manuscript addresses an important methodological framework in algal transcriptomics by systematically evaluating RNA extraction and rRNA depletion protocols across a diverse set of seaweed species. The manuscript is generally well structured and the experimental design appears rigorous. However, there are areas where clarifications are needed before the manuscript can be accepted.

We thank the reviewer for carefully reading our manuscript and for offering valuable suggestions and corrections, which we believe have substantially enhanced the manuscript.

Introduction

Pg. 2, Line No.30-32: The authors stated that “advancing our understanding of seaweed biology, including its response to environmental stressors and pathogens, is essential for enhancing cultivation practices.” Please clarify which stressors (temperature, salinity, pollutants) have already been studied in seaweeds, and which remain underexplored?

We acknowledge that, in hindsight, the introductory paragraph did not adequately represent the current state of research across the different areas. It was not our intention to suggest that little research has been done on seaweed physiology and environmental stressors; on the contrary, this field is well established. Rather, our point was that studies employing genomics and transcriptomics approaches (especially to investigate pathogen interactions) remain relatively limited. We have therefore revised the text to remove any such unintended implications and to provide a more accurate and concise overview of some relevant research fields (L30-35).

Pg. 2, Line No.39-44: The authors have mentioned “RNA-seq limitations in capturing non-polyadenylated RNAs. Since many algal viruses and prokaryotic RNAs lack polyA tails, could the authors provide evidence from prior algal/seaweed transcriptome studies where viral transcripts were missed due to polyA-based methods?

Thank you. To our knowledge, direct demonstrations in literature of “missed entirely” cases in seaweeds are not available. We have revised the text to avoid implying this and now reference algal metatranscriptome studies showing that poly(A)-selected libraries recover substantially fewer viral reads and viral contigs compared to ribodepleted total RNA libraries (L42-48). Briefly, in a brown-tide algal bloom, reads mapping to the Aureococcus anophagefferens virus were ~2 orders of magnitude lower with poly(A) selection than with ribodepletion; the authors therefore recommend rRNA-reduced approaches for community-level virus analyses (Gann et al.). In addition, recent algal virus discovery studies have employed total-RNA/rRNA-depleted metatranscriptomes rather than poly(A) selection, consistent with the need to detect non-polyadenylated viral transcripts (Charon et al.).

Pg. 2, Line No.45–61: Have there been any comparative evaluations of these methods specifically in seaweeds before? If not, please make clear that this study is addressing the first comparison report.

We identified seven studies that focused on optimizing RNA isolation for individual seaweed species or color groups. One additional study evaluated total RNA extraction across multiple seaweed taxa, but it compared only two protocols (CTAB- and SDS-based). From the eight referenced studies, we selected a representative set of protocols, excluding those with only minor variations from methods already included. In addition, to our knowledge, no systematic comparisons of ribodepletion strategies in seaweeds have been published to date, and no commercial ribodepletion kits are currently designed specifically for algal samples. We updated the introduction to outline our protocol selection rationale and to underscore that a more extensive selection of protocols coupled

---

## [Decision Letter · Decision Letter 1]

14 Dec 2025

Evaluation of RNA extraction and rRNA depletion protocols for RNA-Seq in eleven edible seaweed species from brown, red, and green algae

PONE-D-25-34674R1

Dear Dr. Dekker,

We’re pleased to inform you that your manuscript has been judged scientifically suitable for publication and will be formally accepted for publication once it meets all outstanding technical requirements.

Kind regards,

Inbakandan Dhinakarasamy, Ph.D

Academic Editor

PLOS One

Additional Editor Comments (optional):

Authors have satisfactorily addressed all the comments raised in the initial review, and the revisions have improved the overall clarity and quality of the manuscript. The study remains relevant and well executed, providing useful methodological guidance for algal transcriptomics. As all comments have been addressed, the manuscript, in its current form, may be considered for publication.

Additional note (L303–308):

Please consider briefly mentioning that seasonal variation, disease incidence, and pollution levels can influence RNA quality, as acknowledging these factors would further strengthen the discussion.

Reviewers' comments:

Reviewer's Responses to Questions

**Comments to the Author**

Reviewer #1: All comments have been addressed

Reviewer #3: All comments have been addressed

Reviewer #4: All comments have been addressed

2. Is the manuscript technically sound, and do the data support the conclusions?

Reviewer #1: Yes

Reviewer #3: Yes

Reviewer #4: Yes

3. Has the statistical analysis been performed appropriately and rigorously?

Reviewer #1: Yes

Reviewer #3: Yes

Reviewer #4: No

4. Have the authors made all data underlying the findings in their manuscript fully available?

Reviewer #1: Yes

Reviewer #3: Yes

Reviewer #4: Yes

5. Is the manuscript presented in an intelligible fashion and written in standard English?

Reviewer #1: Yes

Reviewer #3: Yes

Reviewer #4: Yes

Reviewer #1: I am pleased to confirm that the authors have fully and satisfactorily addressed all the concerns and suggestions I raised in my initial review. The revisions have significantly improved the clarity, rigor, and overall impact of the manuscript.

The study remains well-designed, systematically executed, and highly relevant to the field of algal transcriptomics. The comparative framework across diverse seaweed species and multiple RNA handling protocols provides much - needed practical guidance for researchers working with non-model macroalgae. It makes a valuable contribution to the methodological literature and will serve as a helpful resource for advancing seaweed genomics and transcriptomics.

1. Line number: L303-308 Seasonal wise variation, disease prevalence pattern and pollution parameter may impact the quality of RNA. It may be included in the manuscript.

Reviewer #3: (No Response)

Reviewer #4: All the clarifications and comments are well justified and modified by the authors. The manuscript may be accepted for publication in the present format

**Do you want your identity to be public for this peer review?** For information about this choice, including consent withdrawal, please see our Privacy Policy

Reviewer #1: **Yes: ** Dr STALIN DHAS T, Assistant Professor (Research), National Facility for Coastal and Marine Research, Sathyabama Institute of Science and Technology, Chennai 600119, Taml Nadu, India

Reviewer #3: **Yes: ** Dr. Jeyapragash Danaraj

Reviewer #4: No

---

## [Editor Report · Acceptance letter]

PONE-D-25-34674R1

PLOS One

Dear Dr. Dekker,

I'm pleased to inform you that your manuscript has been deemed suitable for publication in PLOS One. Congratulations! Your manuscript is now being handed over to our production team.

Kind regards,

on behalf of

Dr. Inbakandan Dhinakarasamy

Academic Editor

PLOS One